# Rationally Designed Pyrimidine Compounds: Promising Novel Antibiotics for the Treatment of *Staphylococcus aureus*-Associated Bovine Mastitis

**DOI:** 10.3390/antibiotics12081344

**Published:** 2023-08-21

**Authors:** Guillaume Millette, Evelyne Lacasse, Renaud Binette, Véronique Belley, Louis-Philippe Chaumont, Céline Ster, Francis Beaudry, Kumaraswamy Boyapelly, Pierre-Luc Boudreault, François Malouin

**Affiliations:** 1Department of Biology, Institute of Sciences, University of Sherbrooke, Sherbrooke, QC J1K 2R1, Canada; guillaume.millette@usherbrooke.ca (G.M.); evelyne.lacasse@usherbrooke.ca (E.L.); celine.ster@agr.gc.ca (C.S.); 2Department of Chemistry, Institute of Sciences, University of Sherbrooke, Sherbrooke, QC J1K 2R1, Canada; renaud.binette@usherbrooke.ca; 3Department of Veterinary Biomedicine, Institute of Veterinary Medicine, University of Montreal, St-Hyacinthe, QC J2S 2M2, Canada; francis.beaudry@umontreal.ca; 4Department of Pharmacology and Physiology, Institute of Medicine and Health Sciences, Sherbrooke University, Sherbrooke, QC J1H 5N4, Canada

**Keywords:** *Staphylococcus aureus*, bovine mastitis, pyrimidine antibiotic, one health

## Abstract

*Staphylococcus aureus* is one of the major pathogens causing bovine mastitis, and antibiotic treatment is most often inefficient due to its virulence and antibiotic-resistance attributes. The development of new antibiotics for veterinary use should account for the One Health concept, in which humans, animals, and environmental wellbeing are all interconnected. *S. aureus* can infect cattle and humans alike and antibiotic resistance can impact both if the same classes of antibiotics are used. New effective antibiotic classes against *S. aureus* are thus needed in dairy farms. We previously described PC1 as a novel antibiotic, which binds the *S. aureus* guanine riboswitch and interrupts transcription of essential GMP synthesis genes. However, chemical instability of PC1 hindered its development, evaluation, and commercialization. Novel PC1 analogs with improved stability have now been rationally designed and synthesized, and their in vitro and in vivo activities have been evaluated. One of these novel compounds, PC206, remains stable in solution and demonstrates specific narrow-spectrum activity against *S. aureus*. It is active against biofilm-embedded *S. aureus*, its cytotoxicity profile is adequate, and in vivo tests in mice and cows show that it is effective and well tolerated. PC206 and structural analogs represent a promising new antibiotic class to treat *S. aureus*-induced bovine mastitis.

## 1. Introduction

Antimicrobial resistance (AMR) is a worldwide public health crisis: there were approximately 5 million human deaths associated with AMR in 2019 [1]. While there are new antibiotics currently being developed, most of these compounds belong to already existing antibiotic classes [2]. Therefore, resistance against these compounds in development will be readily acquired as resistance genes for those antimicrobial classes already exist and are ubiquitous. As a result, new drug classes are much needed [3]. 

One Health is a multidisciplinary approach that addresses conjointly the well-being of humans, livestock, and the environment as a whole [4]. In that context, AMR is not limited to human clinical settings but extends to various other environments. Indeed, wherever antibiotics are used, resistance appears, including in microbes affecting domestic animals and livestock. Furthermore, resistant bacteria and their resistance genetic determinants easily transfer from one reservoir to another [5]. Therefore, it is paramount to address the global AMR crisis by acting in all the different sectors where antibiotics are used. 

In support of the One Health approach, governments worldwide have started to establish policies limiting antibiotic use in animal production [6,7]. Notably, these regulations aim to protect class I antibiotics, i.e., those that are critical for clinical medicine, such as fluoroquinolones and third-generation cephalosporins [8]. Ceftiofur is an example of the latter, which was often used to treat bovine mastitis (i.e., infectious inflammatory disease of the udder) [9]. *Staphylococcus aureus* is the most prevalent bacterial pathogen causing bovine mastitis [10], and new alternatives are immediately needed.

Reported cure rates for *S. aureus* intramammary infections are usually low and depend in part on the genetic background and the virulence factors of *S. aureus* isolates and their ability to produce biofilm [11,12], a protective layer surrounding the bacterial population that limits antibiotic and host defenses to reach and act on their targets [13]. An important consideration with this pathogen is its capacity to infect both human and livestock, leading to an increased risk of emergence of antibiotic-resistant clones like methicillin-resistant *S. aureus* (MRSA) [14,15]. MRSA are frequent in the hospital setting (HA-MRSA) and increasingly so in the community (CA-MRSA), as well as among livestock and animal food production (LA-MRSA). MRSA were classified as “high-priority” on the 2017 WHO’s list of bacteria for which new antibiotics are urgently needed [16].

Identification of bacterial genes expressed during infections can allow the discovery of promising new antibiotic targets. Previously, our group reported that the *S. aureus* gene *guaA* was overexpressed during bovine mastitis [17]. It encodes a guanosine monophosphate (GMP) synthetase and is part of an operon (*xpt-pbuX-guaB-guaA*) controlled by a guanine riboswitch. Riboswitches are structures of noncoding RNA situated in the 5′ untranslated regions of an mRNA molecule [18,19]. In the absence of the guanine ligand, the riboswitch adopts a conformation enabling the transcription of the downstream operon, including *guaA*. When guanine is in a sufficient concentration in the bacteria, it binds to the riboswitch and switches it “off”, stopping the transcription of *guaA* and GMP synthesis. Using rational design, PC1 (Pyrimidine Compound 1; 2,5,6-triaminopyrimidine- 4-one) was found to be structurally similar to guanine and able to block *guaA* expression and subsequent GMP synthesis [18]. PC1 was characterized in vitro and was also shown to significantly reduce *S. aureus* infection in experimental models of intramammary infection in mice and cows [18,20]. PC1 has been demonstrated to be specific toward *S. aureus* and some other staphylococcal species (*e.g.*, *S. epidermidis* and *S. hominis*). This riboswitch is also found in different species, like *Bacillus* sp., but only regulates *guaA* in staphylococci. Unfortunately, PC1 was found to be chemically unstable and to self-dimerize in the presence of water and oxygen, requiring the addition of an antioxidant to remain active. 

In our efforts to engineer chemically stable analogs of PC1, we have discovered PC2, a closely related analogue of PC1 that differs by the absence of the amine at position 5 on the pyrimidine cycle, which is responsible for dimerization (Figure 1). PC2 is more chemically stable and has been shown to bind the riboswitch ex vivo, but possesses much lower antimicrobial activity against *S. aureus* [18]. Using rational design, we synthetized structural analogs of PC2 that are chemically stable and that regained activity against *S. aureus*. In this study, we characterized the best candidate to date, PC206, and three other analogs (PC371, PC372 and PC383), which differ by the length of their alkyl chain bound to the amino group of position 6 on the pyrimidine cycle. The antimicrobial activity of PC206 was evaluated in vitro against *S. aureus* strains from different sources (bovine and human) and the in vitro cytotoxicity was characterized. Furthermore, the efficiency of PC206 was tested in a *S. aureus* mastitis mouse model and initial pharmacokinetics were measured in the udder of dairy cows. Although the mechanism of action of our new pyrimidine analogs remains to be fully elucidated, it represents the first step for the introduction of a new class of antibiotics to treat *S. aureus* bovine mastitis.

## 2. Results

### 2.1. In Vitro Antibacterial Activity

The antibacterial spectrum of activity of novel PC2 analogs (Figure 1) is presented in Table 1. PC206 was the most potent PC2 analog (MIC ranging from 4 to 8 µg/mL) against all reference staphylococci strains tested, including MRSA. All PC2 analogs were less active than PC1 but did not require the addition of an antioxidant in susceptibility tests like PC1. Indeed, we have previously demonstrated that an antioxidant such as dithiothreitol prevents PC1 dimerization and consequently prevents the loss of antibiotic activity [18]. PC372 was also effective against staphylococci strains, with MIC ranging from 8 to 16 µg/mL. It is worth noting that *Enterococcus faecalis* and *Escherichia coli*, including the hyperpermeable *E. coli* strain Δ*lptD*4213, which are bacterial species that are not affected by PC1 and that were used to show specificity [18], were accordingly not significantly affected by these two novel compounds (MIC 128 and >128 µg/mL). All tested bacterial strains were susceptible to PC383, including non-staphylococci strains (MIC ranging from 4 to 16 µg/mL). The lack of specificity of PC383 may be due to its longer lateral carbon chain (10 carbons rather than 5 or 6 for PC372 and PC206, respectively). In contrast, PC371 showed no measurable activity against any of the bacteria tested (MIC > 128 µg/mL); hence, inversely, it is possible that its shorter carbon chain (3 carbons) does not allow it to penetrate the bacterial membrane [21]. 

Because of its efficacy and specificity against *S. aureus*, the antibacterial activity of PC206 was further profiled using isolates from human and animal origins (Table 2). PC206 MIC_50_ and MIC_90_ were 8 and 16 µg/mL, respectively, for both *S. aureus* bovine strains (*n* = 48) and clinical MRSA strains (*n* = 43). For *S. epidermidis* (*n* = 10), the MIC_50_ and MIC_90_ were slightly lower than for the *S. aureus* strains (4 and 8 µg/mL, respectively). For *S. hominis* (*n* = 10), values were 8 and 32 µg/mL for MIC_50_ and MIC_90_, respectively. Unexpectedly, *S*. *haemolyticus*, *S. chromogenes*, and *S. simulans* were much less susceptible (MIC_50_ of 32–64 µg/mL; MIC_90_ of 128 µg/mL), even though the molecular target of the antibiotic should be present in those species. Overall, PC206 showed good MIC_50_ and MIC_90_ against both *S. aureus* bovine strains and MRSA strains and retained specificity for the targeted bacteria.

The kill kinetics of PC206 was studied in vitro against different susceptible staphylococci strains. PC206 slowly killed, in a dose-dependent manner, *S. aureus* Newbould (Figure 2a), *S. aureus* CMRSA-10 (USA-300) (Figure 2b), and *S. epidermidis* ATCC 12228 (Figure 2c). At the highest concentration of PC206 tested (4× MIC), the bacterial viable counts at 24 h were either unaffected or slightly reduced (0.5–1.5 log_10_ CFU/mL) compared to the initial inoculum, thus qualifying PC206 as a bacteriostatic agent.

### 2.2. Antibacterial Activity against Preformed Biofilm

The activity of PC206 was evaluated against *S. aureus* embedded in preformed biofilms on peg lids (Figure 3). PC206 significantly reduced viable counts of the high-producing biofilm *S. aureus* bovine strain 2117 in a dose-dependent manner after 24 h exposure, with a reduction in CFU/peg of 1.28, 2.40, and 5.43 log_10_ compared to the untreated biofilm at concentrations of 32, 128, and 512 µg/mL, respectively. Gentamicin and vancomycin were used as positive and negative controls and results were as previously reported [22]. Gentamicin significantly reduced CFU/peg at a concentration as low as 2 µg/mL, while vancomycin did not impact viable counts of *S. aureus* at concentrations up to 512 µg/mL.

### 2.3. Evaluation of the Binding of Antibiotics to the Riboswitch in a lacZ-Gene Reporter Assay

The novel pyrimidine compounds, especially PC206, display specificity and potency against some staphylococci but not the other bacterial genus (Table 1 and Table 2). Because they are structurally derived from PC1 and PC2, these novel antibiotic candidates are expected to have the same mode of action [18]. Allegedly, these novel antimicrobials should be specific to species possessing a guanine riboswitch controlling *guaA* expression, e.g., the staphylococci. To verify this hypothesis, a strain of *B. subtilis* with a guanine riboswitch-*lacZ* fusion construct (named *xpt-lacZ*) integrated in its chromosome was exposed to these new antimicrobials. Indeed, the guanine riboswitches naturally present in *B. subtilis* do not control the expression of *guaA*, and as a result, *B. subtilis* growth is not significantly affected by antibiotics targeting the guanine riboswitch (MIC shown in Table 1). Therefore, it is an adequate reporter species. The metabolization of substrate ONPG into colored ONP was normalized by the bacterial suspension absorbance at 600 nm (Miller units) and relativized to the untreated condition.

As hoped, PC206 decreased the expression of *lacZ* in the reporter strain in a dose-dependent manner; at the highest evaluated concentration (64 µg/mL), only 35% of the untreated condition signal remained (Figure 4a). PC372 also significantly decreased the expression of Miller units (down to 50%) when applied at a high dose (64 µg/mL) but was overall less effective than PC206 (Figure 4b). This observation is coherent with its higher MIC against *S. aureus* compared to PC206 (16 µg/mL and 8 µg/mL, respectively, Table 1). PC383 did not impact Miller units’ expression (Figure 4c), although the highest evaluated dose (2 µg/mL) was lower than for PC206 and PC372. Indeed, PC383 at higher doses did not allow sufficient growth of the reporter strain to quantify Miller units. This is unsurprising, as its MIC against *B. subtilis* was 4 µg/mL (Table 1). PC371 was also tested and did not have any effect on the reporter gene expression (Appendix A), possibly due to its lack of permeation through the membrane to reach the target [21]. Guanine, the natural ligand of the riboswitch, was used as a positive control and strongly inhibited *lacZ* expression even at a concentration as low as 1 µg/mL (Figure 4d). Oxacillin, which does not bind to the riboswitch, was used as a negative control, and did not affect the expression of Miller units (Figure 4e). Higher concentrations of oxacillin could not be tested as they killed the reporter strain (MIC of 0.25 µg/mL; Table 1). Overall, PC206 inhibits *lacZ* expression in a gene reporter assay, which suggests binding to the guanine riboswitch.

### 2.4. Evaluation of Cytotoxicity (CC_50_)

CC_50_ of PC2 analogs were measured using two different cell lines, MAC-T and HepG2 (bovine mammary epithelial cells and human carcinoma hepatocellular cells, respectively), in two different assays, MTT and LDH release. The MTT assay is based on the cleavage of the yellow tetrazolium salt MTT to form purple formazan crystals by metabolically active cells. Furthermore, the LDH assay is a colorimetric assay for the quantification of cell death and cell lysis, based on the measurement of lactate dehydrogenase (LDH) activity released from the cytosol of damaged cells into the supernatant. These assays are complementary and when used together may provide clues on the mechanism of cytotoxicity. NBD-Cl (4-chloro-7-nitrobenzofurazan) was used as a control because it provides measurable CC_50_ after 24 h and is routinely used in our laboratory for cytotoxicity measures. This control reduces cells viability by inhibiting the ATP production of eukaryotic cells, including human and bovine cells [23,24]. 

All CC_50_ values are reported in Table 3, and plots are provided in the Supplementary Information (Appendix A). PC371 had no effect on cell viability even at the highest concentrations tested (CC_50_ > 600 μM) in both the MTT and the LDH assay and for the two cell lines tested. For the three other compounds, in both cell lines and both assays, a correlation was observed between the effect on cell viability and the length of their aliphatic chain. CC_50_ of PC372 (5 carbons) was higher than that of PC206 (6 carbons), and both were higher than that of PC383 (10 carbons). Indeed, for the MAC-T cells in the MTT assay, the values were >600 μM (PC372), 317 μM (PC206), and 13.5 μM (PC383). Again, for the MAC-T cell line, the values for the LDH assay were >600 μM (PC372 and PC206) and 23.6 μM (PC383). Similar results were obtained with the HepG2 cells. CC_50_ values measured with the MTT assay were 341 μM (PC372), 160 μM (PC206), and 11.7 μM, (PC383) and the CC_50_ values obtained with the LDH assay were 394 μM, 149 μM, and 6.6 μM, respectively. Overall, the MAC-T cells tolerated better exposure to the compounds than the HepG2 cells. To assess the selectivity of the PC2 analogs towards *S. aureus*, selectivity ratios were calculated. These ratios were obtained by dividing the cytotoxicity CC_50_ values of an analog by its MIC against *S. aureus* Newbould. It was not possible to calculate the selectivity ratio for PC371 as its CC_50_ and MIC values exceeded the maximum values tested. In contrast, PC383 exhibited no selectivity as its ratio was close to 1, indicating its inability to differentiate between *S. aureus* and eukaryotic cells. Both PC372 and PC206 were largely more selective than PC383, consistently yielding selectivity values above 3.9.

### 2.5. Efficacy of PC206 in a Mouse Mastitis Model

The efficacy of PC206 was assessed in a well-established mouse mastitis model of *S. aureus* intramammary infection [25]. Briefly, mice were anesthetized, and mammary glands were inoculated with *S. aureus* Newbould. The mice were then either given two intramammary doses of 500 µg of PC206, or the equivalent volume of excipient, 1 and 4 h after infection. Mice were euthanized 10 h after the last injection (14 h post-infection), mammary glands were harvested, and bacterial loads in the glands were measured. No inflammation, damage, nor adverse effects were observed in the animals or the mammary gland tissues following administration of PC206. The mice treated with PC206 had significantly lower bacterial counts than untreated mice, with a difference in median bacterial counts of 1.24 log_10_ CFU/g of gland (Figure 5). Therefore, PC206 demonstrated significant antibacterial activity in the mouse mastitis model of *S. aureus* intramammary infection. 

### 2.6. Metabolic Stability and Initial Safety and Pharmacokinetics Assessments in Cows

Before any experimentation in cows, mouse metabolic and plasmatic stability of PC206 was assessed over a period of 60 min (Appendix A). The stability of new chemical entities to metabolic biodegradation is an important factor when evaluating the possibility of developing compounds into new drugs. Metabolism is a key contributor to drug clearance and directly influences systemic drug exposure. PC206 metabolic stability was evaluated in CD-1 female mice liver S9 fraction containing drug-metabolizing enzymes. The data presented in Appendix A were fitted with a mono-exponential decay model. PC206′s half-life value (T_1/2_) was determined to be 83.3 min, indicating that PC206 is not rapidly metabolized. Consequently, if systemic exposure is eventually intended for PC206, the duration of action will not be impaired by rapid clearance. Moreover, plasma stability was performed and PC206 was stable. Next, in a preliminary safety assessment study in cows, three single doses of PC206 (100, 250, and 500 mg), or saline as a control, were directly infused in the four quarters of four healthy cows immediately after morning milking, as this is carried out with other commercially available drugs for the treatment of bovine mastitis. The somatic cell counts and quarter-milk production remained unaffected by the treatments during the 69 h following administration by infusion of PC206, indicating no induction of inflammation by the compound (Appendix A). Body temperature was also unaffected (maintained around 38 °C for every cow). Neither redness nor swelling was observed at the time of milking in any of the quarters. Considering the pharmacokinetics of PC206 following treatment, its recovery in quarter milk was dose-dependent. With doses of 100, 250, and 500 mg, the concentration of PC206 remained above the MIC for *S. aureus* mastitis strains (MIC_50_ 8 µg/mL; Table 2) over 2, 4, and for about 8 h, respectively (Figure 6).

## 3. Discussion

Treatments for *S. aureus* bovine mastitis are limited or not efficient [12,26], and the use of third- and fourth-generation cephalosporins in dairy farms is a concern from a One Health perspective [12,26]. New specific treatments are clearly needed. We have previously reported PC1, a pyrimidine compound binding the guanine riboswitch of *S. aureus* and inhibiting the expression of the essential gene *guaA* [18]. However, PC1 tends to self-dimerize in the presence of water and oxygen and therefore requires the addition of an antioxidant adjuvant, i.e., dithiothreitol, to maintain its antibiotic activity. By removing the amino group at position 5 on the pyrimidine cycle of PC1, we obtained PC2, which is chemically stable and able to bind the riboswitch ex vivo but loses most of its activity against *S. aureus* [18]. We hypothesized that the lack of antibacterial activity of PC2 was due to its inability to penetrate the bacterial membrane passively due to its high hydrophilicity, whereas analogs with a hydrophobic aliphatic chain could. Therefore, we set out to develop structural analogs of PC2 that would remain chemically stable and regain activity against *S. aureus*.

In this article, we have characterized four analogs of PC2 with aliphatic chain of various lengths on the amino group at position 6 on the pyrimidine cycle. PC372 (5-carbon chain) and PC206 (6-carbon chain) displayed specific antibacterial activity against different staphylococci species. Accordingly, PC371, with its shorter 3-carbon chain, did not show any activity. The alkyl chain of this compound may not possess sufficient lipophilicity to significantly decrease the polarity necessary to enable its passive diffusion through bacterial membranes [21]. Conversely, PC383 (10-carbon chain) was unspecific and targeted staphylococci and non-targeted species (i.e., *E. coli* and *E. faecalis*) alike. PC383 was also shown to be cytotoxic in vitro for HepG2 and MAC-T cells, possibly due to its longer carbon chain causing non-specific cytotoxicity and cell damage through its interactions with cell membranes (i.e., very low CC_50_ value in the LDH assay against HepG2 cells; Table 3). Overall, there seems to be a balance point for the ideal length of the alkyl chain. If it is too short, the drug will not be able to cross the bacterial membrane; if it is too long, it will lose its specificity and cause damage to all cells.

PC206 was selected as the most promising of the four analogs because of its selective antimicrobial activity against staphylococci. It was also shown to remain active in biofilm, which is a challenge in antibiotic development [13]. PC206 has low and equivalent MIC_50_ and MIC_90_ against a panel of *S. aureus* bovine strains and MRSA strains. On the other hand, coagulase-negative staphylococci (CNS) were not equally susceptible to the drug; *S. hominis* and *S. epidermidis* displayed similar MIC_50_ and MIC_90_ values to *S. aureus*, while *S. chromogenes*, *S. haemolyticus*, and *S. simulans* were not as susceptible. A possible explanation for the reduced activity of PC206 against some CNS could be differences in their membrane composition, such that PC206 polarity would be adequate for certain species but not for others [27]. Also, while PC206 is derived from PC1, we have not investigated his mode of action as thoroughly as for PC1. Further studies are needed to fully understand the antibiotic mode of action of these novel pyrimidine compounds. Interestingly, the lack of suitable antibiotic activity of PC206 against some CNS might be beneficial as it was previously demonstrated that *S. chromogenes*, a CNS, has a protective effect against *S. aureus* in a mouse co-infection model [28].

The mode of action of our novel pyrimidine compounds was nonetheless explored to ascertain if there were indeed similarities with the antibiotic class representatives PC1 and PC2. Hence, we sought to verify if PC206 could bind to the guanine riboswitch, like its structural precursors, using a guanine riboswitch-*lacZ* fusion construct integrated in *B. subtilis* [18]. Both *S. aureus* and *B. subtilis* belong to the order of the *Bacillales* and have the same essential nucleotides for the binding of guanine to their respective riboswitch [18]. However, the riboswitch in *B. subtilis* is upstream of *xpt* and *pbuX*, while in *S. aureus*, it also precedes essential genes *guaA* and *guaB*, thus tentatively explaining its differential antibiotic action. Therefore, we considered our model to accurately assess the binding of molecules to the riboswitch when β-galactosidase activity is reduced in the presence of a riboswitch ligand. In our model, PC206 and, to a lesser degree PC372, the only two compounds that showed specific antibiotic activities toward staphylococci, significantly decreased in Miller units in comparison to the untreated control (Figure 4). PC371 did not show any detectable binding affinity as it probably could not penetrate through the *B. subtilis* cytoplasmic membrane. PC383, which has unspecific antibiotic activity and displayed the highest toxicity against eukaryotic cell lines, did not impact the LacZ signal even at concentrations that were toxic for the reporter strain. Here, the binding of these novel pyrimidine compounds to the riboswitch was specifically studied because PC1 has previously been shown to bind to the guanine riboswitch and inhibit the expression of the genes under its regulation [18]. However, the validity of the riboswitch as an antibiotic target has since been questioned in some experimental conditions [29]. It is thus possible that there is another or an additional antibiotic target other than the guanine riboswitch, despite the correlation between MIC and the reduction in *lacZ* expression in the β-galactosidase assay. Nevertheless, PC206 is a very narrow-spectrum antibiotic of a novel structural class, which is only effective against some staphylococci. 

The innocuity of our novel pyrimidine compounds was verified for two eukaryotic cell lines, HepG2 and MAC-T, with two different assays, LDH and MTT. PC206 displayed little to no cytotoxicity, depending on the cell line and the assay used. Its selectivity index (S.I.), i.e., cytotoxicity compared to its antibiotic activity, is comparable to PC372, both of which are the most selective PC2 analogs. The S.I. allows us to compare antibiotics among themselves, but it does not warrant good tolerability in animals. PC206 cytotoxicity (for MAC-T cells, CC_50_ of 317 and >600 μM [66 and >125 μg/mL], respectively for MTT and LDH assays; Table 3) is comparable to antibiotics targeting *S. aureus* in bovine mastitis. For example, cytotoxicity CC_50_ values reported for enrofloxacin and ikarugamycin in MAC-T cells were 345 and 9.2 μg/mL, respectively (871 and 19 μM, respectively) [30,31]. The evaluation of cytotoxicity on cell monolayers does not always correlate well with in vivo tolerability due to many factors, including the cell line, the antibiotic contact time, and the assay conditions [32,33]. Therefore, the innocuity of PC206 was further verified in dairy cows. When injected into the mammary gland quarters, PC206 did not impact somatic cells count, body temperature, milk production, and appearance, nor did it cause any visible symptoms during the 69 h observation period. 

The in vivo efficacy of PC206 was demonstrated using a well-established murine infectious mastitis model [25]. The administration of PC206 reduced viable counts of *S. aureus* Newbould by 1.24 log_10_ CFU/g of gland, which was statistically significant. Mouse hepatic and plasmatic stability of PC206 was also assessed and was relatively stable in both environments over the course of the experiment (Appendix A). We then measured the recovery of PC206 in quarter milk of cows after a single intramammary infusion. The concentration of PC206 in the milk exceeded the MIC for *S. aureus* for nearly 8 h when used at the highest dose (500 mg). In a future study, PC206 in vivo activity, pharmacokinetics, and pharmacodynamic properties could be further explored, to optimize dosing regimens, thereby improving treatment outcomes. For example, an appropriate formulation could be tested to fit dairy farm practices that often require milking every ~12 h. It should also be noted that intramammary infusion is the usual administration procedure for anti-mastitis therapies and that ceftiofur can be used at a dose of 125 mg/quarter every 24 h for up to 8 days or cephapirin at a dose of 250 mg every 12 h. Hence, our infusion procedure and dosing proposition for PC206 is already similar to dairy farm standards. 

In conclusion, although its mode of action remains to be completely elucidated, this study proposes PC206 as the first stable candidate of a novel antibiotic class dedicated to the treatment of *S. aureus*-induced mastitis in dairy cows. Its potency and specificity were confirmed with MIC assays on large panels of *S. aureus* isolates from human and animal origins and against preformed biofilms. Furthermore, its impact on eukaryotic cells was thoroughly examined and deemed suitable. Finally, PC206 safety, effectiveness, and pharmacokinetics were evaluated in vivo, including in a mouse mastitis model and in lactating cows. With one-step synthesis using inexpensive, commercially available material, we expect a low-cost production, which is critical for use in food-animal production. This should help reduce broad-spectrum antibiotic overuse in farm animals, allowing clinically important antimicrobials for human health care to be preserved. 

## 4. Materials and Methods

### 4.1. Synthesis of Novel Pyrimidine Analogs of PC1

This study specifically focused on novel pyrimidine analogs that have an amino group with an alkyl chain of varying lengths at position 6 (Figure 1a). Alkylated PC2 analogs were synthesized using the same protocol with a nucleophilic aromatic substitution at position 6 from commercially available 6-chloropyrimidine-2,4-diamine with the appropriate alkylamine (Figure 1b). The products were then purified by Prep-HPLC- MS. Syntheses are described below, and characterization data are provided in the Supplementary Information (Appendix A). 

In a 2 mL glass vial, alkylamine (0.5 mL) was added dropwise to solid 6-chloropyrimidine-2,4-diamine (50 mg, 0.34 mmol) before adding a 7 mm × 2 mm magnetic stir bar. The vial was sealed, and the reaction mixture was stirred at 110 °C for 24 h. The crude mixture was diluted with methanol to a total volume of 2 mL, filtered, and directly injected for purification. Purification was carried out on a Waters preparative HPLC-MS (column XSelect^TM^ CSH^TM^ Prep C18 (19 × 100 mm) packed with 5 μm particles, UV detector 2998, MS SQ Detector 2, Sample manager 2767, and binary gradient module). The eluents used were acetonitrile and water, both containing 0.1% formic acid.

A first run was conducted with the above eluents and the following method: 5% ACN to 95% ACN (0 → 10 min) in order to assess the optimal %ACN for product elution. A second run was made using the following eluent: [(%ACN) − 10 → (%ACN) + 5] (0 → 15 min). Fractions collected were analyzed on a Waters UPLC-MS system (column Acquity UPLC CSH^TM^ C18 (2.1 mm × 50 mm) packed with 1.7 μm particles) using ACN and water + 0.1% formic acid. The method used was 0 → 0.2 min: 5% ACN; 0.2 → 1.5 min: 5% → 95% ACN; 1.5 → 1.8 min: 95% ACN; 1.8 → 2.0 min: 95% → 5% ACN; 2.0 → 2.5 min: 5% ACN. Pure fractions were pooled and lyophilized. All compounds possessed ≥95% purity as determined by UPLC-MS. HRMS spectra of the final compounds were recorded using a maXis ESI-Q-TOF with ESI in positive mode. 

### 4.2. Characterization of Novel Pyrimidine Analogs of PC1

#### 4.2.1. PC206

Yield: 32% (23 mg), off-white fluffy powder; UPLC-MS: retention time of 1.28 min, purity 98.8%, mass found (m/z) [M + 1]^+^: 210.1; ^1^H NMR (400 MHz, DMSO-*d*6) δ ppm 0.84 (t, *J* = 6.50 Hz, 3 H) 1.17–1.33 (m, 6 H) 1.45 (quin, *J* = 7.00 Hz, 2 H) 3.12 (q, *J* = 5.50 Hz, 2 H) 4.98 (s, 1 H) 6.77 (br. s., 2 H) 6.94 (br. s., 2 H) 7.31 (br. s., 1 H); ^13^C NMR (400 MHz, DMSO-*d*6) δ ppm 167.24, 161.78, 156.48, 72.45, 40.58, 31.06, 28.84, 26.15, 22.14, 13.95; HRMS (m/z): [M + H]^+^ calcd. for C_10_H_20_N_5_, 210.1713; found, 210.1717.

#### 4.2.2. PC371

Yield: 15% (9 mg), off-white fluffy powder; UPLC-MS: retention time of 1.05 min, purity 98.1%, mass found (m/z) [M + 1]^+^: 167.9; ^1^H NMR (400 MHz, DMSO-*d*6) δ ppm 0.86 (t, *J* = 7.43 Hz, 3 H) 1.47 (sxt, *J* = 7.28 Hz, 2 H) 3.07 (q, *J* = 6.43 Hz, 2 H) 4.90 (s, 1 H) 6.18 (br. s., 4 H) 6.72 (br. s., 1 H); ^13^C NMR (400 MHz, DMSO-*d*6) δ ppm 165.46, 162.88, 159.20, 73.07, 42.30, 22.28, 11.48; HRMS (m/z): [M + H]^+^ calcd. for C_7_H_14_N_5_, 168.1244; found, 168.1246.

#### 4.2.3. PC372

Yield: 28% (19 mg), off-white fluffy powder; UPLC-MS: retention time of 1.21 min, purity 97.6%, mass found (m/z) [M + 1]^+^: 196.0; ^1^H NMR (400 MHz, DMSO-*d*6) δ ppm 0.87 (t, *J* = 7.00 Hz, 3 H) 1.21–1.33 (m, 4 H) 1.46 (quin, *J* = 7.09 Hz, 2 H) 3.12 (q, *J* = 6.00 Hz, 2 H) 4.95 (s, 1 H) 6.57 (br. s., 2 H) 6.66 (br. s., 2 H) 7.11 (br. s., 1 H); ^13^C NMR (400 MHz, DMSO-*d*6) δ ppm 165.28, 161.09, 155.91, 72.39, 40.53, 28.61, 28.52, 21.90, 13.96; HRMS (m/z): [M + H]^+^ calcd. for C_9_H_18_N_5_, 196.1557; found, 196.1558.

#### 4.2.4. PC383

Yield: 31% (28 mg), off-white fluffy powder; UPLC-MS: retention time of 1.53 min, purity 97.2%, mass found (m/z) [M + 1]^+^: 266.3; ^1^H NMR (400 MHz, DMSO-*d*6) δ ppm 0.85 (t, *J* = 7.00 Hz, 3 H) 1.18–1.32 (m, 14 H) 1.46 (quin, *J* = 6.50 Hz, 2 H) 3.14 (q, *J* = 5.00 Hz, 2 H) 4.99 (s, 1 H) 6.82 (br. s., 2 H) 6.98 (br. s., 2 H) 7.38 (br. s., 1 H); ^13^C NMR (400 MHz, DMSO-*d*6) δ ppm 165.26, 160.95, 155.51, 72.40, 40.57, 31.32, 29.06, 28.98, 28.79, 28.73, 28.44, 26.42, 22.12, 13.99; HRMS (m/z): [M + H]^+^ calcd. for C_14_H_28_N_5_, 266.2339; found, 266.2344.

### 4.3. Bacterial Strains

Reference strains for this study (*S. aureus* ATCC 29213, ATCC 29740 [bovine mastitis strain Newbould], *S. epidermidis* ATCC 12228, *E. coli* ATCC 35695, *E. faecalis* ATCC 29212) were obtained from the American Type Culture Collection (ATCC, Rockville, MD). *S. aureus* strains CMRSA-10 and CMRSA-2 were obtained from the *Laboratoire de santé publique du Québec* (LSPQ, Sainte-Anne-de-Bellevue, QC, Canada) and are Canadian CA-MRSA USA-300 and HA-MRSA USA-100 clones, respectively. The strain *E. coli* imp4213 (*lptD*), used as a control in MIC assays, is a derivative of *E. coli* ATCC 35695 (*E. coli* MC4100) with an increased membrane permeability [34]. This strain possesses a mutation in the *lptD* gene (LPS transport), resulting in an outer-membrane permeability defect and thus hyper-susceptibility to a variety of antibiotics, including vancomycin, which is normally inactive against Gram-negative bacteria due to the permeability barrier [35]. Other MRSA strains used in this study are clinical isolates of human origin obtained from the microbiology laboratory of the *Centre Hospitalier Universitaire de Sherbrooke* (CHUS, Sherbrooke, QC, Canada). Bovine mastitis strains (*S. aureus*, *S. chromogenes*, *S. haemolyticus*, *S. hominis*, *S. simulans*) were obtained from the Canadian Bovine Mastitis and Milk Quality Research Network (CBRQMN, St-Hyacinthe, QC, Canada) [10]. Among these strains, *S. aureus* 2117 is a high producer of biofilm [10] and was used in the biofilm-eradication assay. *S. epidermidis* isolates were provided by *Centre de Recherche en Infectiologie* (CRI) de Québec (Québec, QC, Canada). *Bacillus subtilis xpt-lacZ* had a transcriptional fusion construct integrated in its genome by recombination as previously described and was used in a reporter gene assay [18].

### 4.4. Antibiotic Susceptibility Testing

Susceptibility tests were performed by a broth microdilution assay in 96-well plates according to the Clinical and Laboratory Standards Institute (CLSI) guidelines [36]. Briefly, each compound was serially diluted 2-fold and bacteria were inoculated in approximately 5 × 10^5^ colony-forming units per mL (CFU/mL). Testing of oxacillin against MSSA and MRSA strains and coagulase-negative staphylococci was performed in cation-adjusted Mueller–Hinton (CAMHB) broth, supplemented with 2% NaCl. The supplemental NaCl was excluded for the testing of the other drugs. Every tested strain was incubated for 20 h at 35 °C, except when exposed to vancomycin and oxacillin; strains were then incubated for 24 h at 35 °C. The minimal inhibitory concentration (MIC) of each drug is defined as the lowest concentration of the antibiotic that prevents visible growth of a bacterial strain. It was determined by measuring the absorbance at 600 nm with an Epoch microplate reader (Agilent Technologies, Mississauga, ON, Canada), as well as by visual confirmation. Vancomycin (Sigma Aldrich, Oakville, ON, Canada) was used as a control antibiotic for Gram-positive bacteria, and oxacillin (Sigma Aldrich) was also used as a control for *S. aureus*, MRSA, and the *B. subtilis xpt-lacZ* reporter strain. Ceftazidime (Sigma Aldrich) was used as a control for *E. coli* strains. Pirlimycin (Pfizer, Kirkland, QC, Canada) was included as the mastitis antibiotic control.

The antibacterial activity of the most promising compound, PC206, was also evaluated against larger groups of clinical isolates of human and animal origin. The MIC_50_ and MIC_90_ values represented the concentrations of drug required to inhibit the growth of 50% and 90% of the bacterial strains tested, respectively.

### 4.5. Kill Kinetics

Time-kill kinetics studies were performed to define the bactericidal effect of PC206. *S. aureus* Newbould, CMRSA-10, or *S. epidermidis* ATCC 12228 were inoculated at a density of approximately 5 × 10^5^ CFU/mL in CAMHB in the absence or presence of PC206 and were incubated with agitation (225 rpm) for 24 h at 35 °C. Right before inoculation and 0.5, 1, 2, 4, 8 and 24 h thereafter, bacterial cultures were sampled, serially diluted, and plated on tryptic soya agar (TSA) plates that were incubated at 35 °C for 24 h to determine viable counts with a detection limit of 2 log_10_ CFU/mL. 

### 4.6. Biofilm-Eradication Test

The viability of bacteria in preformed biofilms treated with PC206 was evaluated using the MBEC assay™ system (formerly the Calgary Biofilm Device) [22,37,38]. The device consists of a plastic lid with 96 pegs and a 96-well plate. The wells were first inoculated with a suspension of *S. aureus* bovine strain 2117 adjusted to a 0.5 McFarland standard in brain heart infusion (BHI) media supplemented with 0.25% glucose, and then covered with the peg lids. Plates were incubated at 35 °C for 24 h with agitation (120 rpm). The biofilm formed on the pegs was then washed three times with PBS and was further incubated in a 96-well plate containing BHI, 0.25% glucose, and antibiotics (two-fold dilutions) at 35 °C for 24 h with agitation (120 rpm). The pegs were washed three times with PBS and the biofilm was sonicated for 10 min, and then transferred into a new 96-well base plate. The 96-well plate was centrifuged for 5 min at 180 g to recover the detached bacteria. Bacteria were resuspended, serially diluted, and plated on TSA plates that were incubated at 35 °C for 24 h to determine the remaining viable counts with a detection limit of 10 CFU/mL. Gentamicin and vancomycin were used as positive and negative controls, respectively. 

### 4.7. Riboswitch Regulation Assay

β-galactosidase gene expression in the presence of the new pyrimidine analogs was studied, using an *xpt-lacZ* transcriptional fusion construct integrated in the genome of *B. subtilis* as previously described [18]. Briefly, cultures were grown for 5 h at 35 °C in CAMHB, from an initial optical density of 0.05 at 600 nm (OD_600_), and then diluted 1:20 and in the presence of different concentrations of the compounds tested. Cultures were centrifuged for 10 min at 13,100 rpm, the supernatant was discarded, and pellets were suspended in a Z-buffer (NaH_2_PO_4_ 40 mM, Na_2_HPO_4_ 60 mM, MgSO_4_ 1 mM, KCl 10 mM). Cultures OD_600_ were measured, and bacteria were then lysed with lysozyme (1 mg/mL) and β-mercaptoethanol 50 mM for 30 min. Lysates were centrifuged for 10 min at 13,100 rpm, and supernatants were incubated for 5 min at 30 °C with the substrate orthonitrophenyl-β-galactoside (ONPG; Sigma Aldrich). Reactions were stopped with Na_2_CO_3_ 1M, supernatants OD_420_ were measured, and Miller units were calculated [39]. A decrease in β-galactosidase activity indicated binding of the test compound to the riboswitch. Guanine and oxacillin were used as negative and positive controls, respectively. The formula to measure Miller units is as follows: $$\mathrm{Miller}\mathrm{units}=1000\times \frac{{\mathrm{OD}}_{420}}{t\cdot v\cdot {\mathrm{OD}}_{600}}$$

Here, t represents the time of incubation with ONPG, and v represents the volume of the reaction. 

### 4.8. Evaluation of Cell Viability Using the MTT Assay

A bovine mammary epithelial cell line, MAC-T, and a human hepatocellular cell line, HepG2, were used to evaluate the effect of the various pyrimidine analogs on cell viability. Both cell lines were cultured in Dulbecco’s modified Eagle medium (DMEM) with 4.5 g glucose per liter and supplemented with 10% heat-inactivated fetal bovine serum (FBS), 1% sodium pyruvate, and 1% antimycotic solution. Cell cultures were incubated at 37 °C with 5% CO_2_.

Twenty-four hours prior to the MTT assay, cells were seeded in a 96-well plate at a concentration of 1 × 10^5^ and 2 × 10^5^ cells/mL for MAC-T and HepG2, respectively. The next day, in a separate plate, pyrimidine compounds and 4-chloro-7-nitrobenzofurazan (NBD-Cl) control were serially diluted in the cell culture medium (DMEM containing 1% FSB, 1% sodium pyruvate and 1% antimycotic solution). DMSO was adjusted to 0.1%. The MTT assay was performed in triplicate according to the kit recommendations (Roche, Cell proliferation kit I (MTT)). Briefly, the cells were exposed to various concentrations of pyrimidine compounds. High-viability controls consisted of cells exposed to the cell culture medium with 0.1% DMSO. After 24 h of incubation, 10 µL of the 3-(4,5-dimethylthiazol-2-yl)-2,5-diphenyltetrazolium bromide (MTT) solutions was added directly on the cells and incubated for four more hours. The solubilization buffer was then added and incubated overnight. Absorbances were measured at 550 nm (OD_550_) to detect the formation of formazan and at 690 nm (OD_690_) for reference using an Epoch microplate reader. The OD_550_ was corrected (cOD_550_) by subtracting the OD_690._ Viability was measured using the following formula: $$\%\;\mathrm{viability}=\frac{{\mathrm{cOD}}_{550}}{\mathrm{mean}\;\mathrm{of}\;\mathrm{high}\;\mathrm{viability}\;\mathrm{controls}}\times 100\%$$

For each compound, the cytotoxic concentration 50% (CC_50_), i.e., the concentration leading to 50% cell death, was determined using a non-linear regression fit (variable slope, four parameters) of the plot of the Log_10_ inhibitor concentration versus the percent cell viability with the software GraphPad prism 9.4.1. 

### 4.9. Evaluation of the Cell Viability by the LDH Release Assay

The MAC-T and HepG2 cell lines were routinely cultured as explained in the MTT assay section above.

The lactate dehydrogenase release (LDH) assay was used to verify the impact of the new pyrimidine analogs on cell membrane integrity. Damaged cell membranes will release LDH in the medium, and the enzyme activity can be quantified. Here, cells were seeded in a 96-well plate at concentrations of 5 × 10^4^ and 1 × 10^5^ cells/mL for MAC-T and HepG2, respectively. Antibiotics and controls were serially diluted in a second plate in the same medium as for the MTT assay. DMSO was adjusted to 0.1%. Triton 1% was used as a complete lysis control and wells that did not contain any test compounds were used as high-viability controls. The LDH assay was conducted in triplicate and according to the kit recommendations (Roche, Cytotoxicity detection kit (LDH)). The cells were exposed to various concentrations of test compounds or controls. After 24 h of incubation, 100 µL of the cell supernatant was sampled and used for the quantification of the LDH activity. Cell supernatant was incubated with the revelation buffer for 20 to 30 min in the dark and absorbances were measured at 492 nm (OD_492_) and for reference at 690 nm (OD_690_) using an Epoch microplate reader. The OD_492_ was corrected (cOD_492_) by subtracting the OD_690_. The formula to measure the LDH release is as follows: $$\%\;\mathrm{LDH}\;\mathrm{release}=\frac{{\mathrm{cDO}}_{492}\;-\;L}{H\;-\;L}\times 100\%$$

Here, L represents the mean of the low LDH release (high viability) and H the high LDH release (complete cell lysis). CC_50_ were measured as explained in the MTT assay section above.

### 4.10. Murine Mastitis Model

A well-characterized mouse mastitis model was used to evaluate the in vivo efficacy of PC206 [25]. Briefly, CD-1 lactating mice were separated from their pups (12–14 days following birth), anesthetized using isoflurane, and infected with *S. aureus* Newbould. For inoculation, the fourth pair of glands found from head to tail (L4 and R4 glands) was first disinfected with 70% ethanol. Then, 100 µL of PBS containing 75–125 CFU was slowly injected into the lactiferous duct with a 32-gauge blunt needle attached to a 1 mL syringe. One and four hours post-inoculation, mice were anesthetized again, and 500 µg of compound PC206 (in PBS) was injected directly into the previously infected mammary glands (6 mammary glands: *n* = 6). Some mice were injected with PBS instead and were used as an untreated control group (8 mammary glands: *n* = 8). Each infected gland was considered to be the experimental unit. Then, 14 h after bacterial inoculation, mice were anesthetized and humanely euthanized, and mammary glands were harvested and homogenized. CFU counts were obtained after plating serial dilution of mammary gland homogenates on TSA plates that were incubated at 35 °C for 24 h. The detection limit was approximately 200 CFU per gram of mammary glands. 

### 4.11. Metabolic and Plasmatic Stability

#### 4.11.1. Metabolic Stability 

All incubations were performed minimally in triplicate. A total of 1 mL of mice liver S9 fraction (XenoTech, Kansas City, KS, USA) was fortified with 50 μL of NADPH-regenerating solution A and 10 μL of solution B (Corning BD Biosciences catalog no. 451200) and preincubated at 37 °C for 5 min prior to the addition of PC206. Immediately after fortification of PC206 into the mice liver S9 fraction containing the NADPH-regeneration system, the sampling points were taken at 2, 5, 10, 15, 30, 45 and 60 min for metabolic stability experiments (i.e., the substrate concentration was 10 μM). Samples of 50 μL were taken and mixed with 250 μL of the internal standard solution (caffeine) in microcentrifuge tubes. The samples were centrifuged at 12,000× *g* for 10 min and 200 μL of the supernatant was transferred into an injection vial for HPLC-MS analysis. A Q Exactive Orbitrap Mass Spectrometer was interfaced with a Vanquish Flex UHPLC system (Thermo Fisher Scientific, Rochester, NY, USA). Chromatography was achieved using a gradient mobile phase, along with a microbore column, namely Thermo BioBasic Phenyl (ThermoFisher Scientific), of 50 × 1 mm with a particle size of 5 μm. The initial mobile phase condition consisted of acetonitrile and water (both fortified with 0.1% of formic acid) at a ratio of 5:95. From 0 to 1 min, the ratio was maintained at 5:95. From 1 to 3 min, it was set to a ratio of 80:20. The mobile phase composition ratio was reverted at the initial conditions, and the column was allowed to re-equilibrate for 5 min for a total run time of 8 min. The flow rate was fixed at 75 µL/min and 2 µL of samples was injected. MS detection was performed in positive-ion mode, operating in high-resolution accurate-mass (HRAM) scan mode. Nitrogen was used for sheath and auxiliary gases and was set at 10 and 5 arbitrary units. The ESI probe was set to 4000 V and the ion transfer tube temperature was set to 200 °C. The scan range was set to m/z 200–500. Data were acquired at a resolving power of 140,000 (FWHM) using an automatic gain control target of 1.0 × 10^6^ and maximum ion injection time of 100 msec. Targeted drug quantification was performed by MS detection using specific precursor masses based on monoisotopic masses (i.e., [M + H]^+^ ions). Quantification was performed by extracting specific precursor ions using a 5 ppm mass window. Instrument calibration was performed prior to all analysis and mass accuracy was notably below 1 ppm using Pierce^TM^ LTQ Velos ESI positive-ion calibration solution (ThermoFisher Scientific) and automated instrument protocol.

#### 4.11.2. Plasmatic Stability 

Blood samples were collected from CD-1 female mice by jugular vein puncture and subsequently transferred to K_3_-EDTA-coated tubes (Sarstedt, St-Leonard, QC, Canada). Samples were gently inverted and centrifuged at 1500× *g* for 10 min at 4 °C, and plasma aliquots were stored at −80 °C until analysis. All incubations were performed in triplicate. A total of 1 mL of plasma was fortified with 10 µM of PC206 and incubated at room temperature (~22 °C). Samples were taken at 2, 5, 10, 15, 30, 45, and 60 min. Plasma samples of 50 μL were taken and mixed with 250 μL of the internal standard solution (caffeine) in microcentrifuge tubes. The samples were centrifuged at 12,000× *g* for 10 min, and 200 μL of the supernatant was transferred into an injection vial for HPLC-MS analysis. Drug detection was performed as described in the previous paragraph. 

### 4.12. Initial Safety and Pharmacokinetics Assessments in Cows

The innocuity of PC206 was evaluated in cows, as previously described for other test substances [20]. Following morning milking, each of the four mammary gland quarters four 4 cows was infused with various doses of PC206 (100, 250, 500 mg) or saline as a negative control. Milk samples were collected aseptically from each quarter prior to treatment, as well as 2, 4, 8, 21, 33, 45, and 69 h after intra-mammary infusion of PC206 or saline to assess the onset of inflammation by determining the somatic cell counts through Lactanet (Saint-Anne-de-Bellevue, QC, Canada). Cows were milked using an individual quarter-milking unit in order to measure individual quarter-milk production. PC206 concentration in milk was also measured; milk samples were diluted with an equal volume of isopropanol and washed 3 times with hexane. PC206 was extracted using ethyl acetate, filtered on Na_2_SO_4,_ and evaporated. The dried samples were diluted 80:20 in ACN:H_2_O containing an internal standard. The content of PC206 was determined using single-ion monitoring (SIMS) on UPLC-MS. 

## 5. Patents

The pyrimidine compounds described in this article are the subject of several issued and pending patent applications, including US patent no. 9,993,491 B2 (issued on 12 June 2018).

## Figures and Tables

**Figure 1 antibiotics-12-01344-f001:**
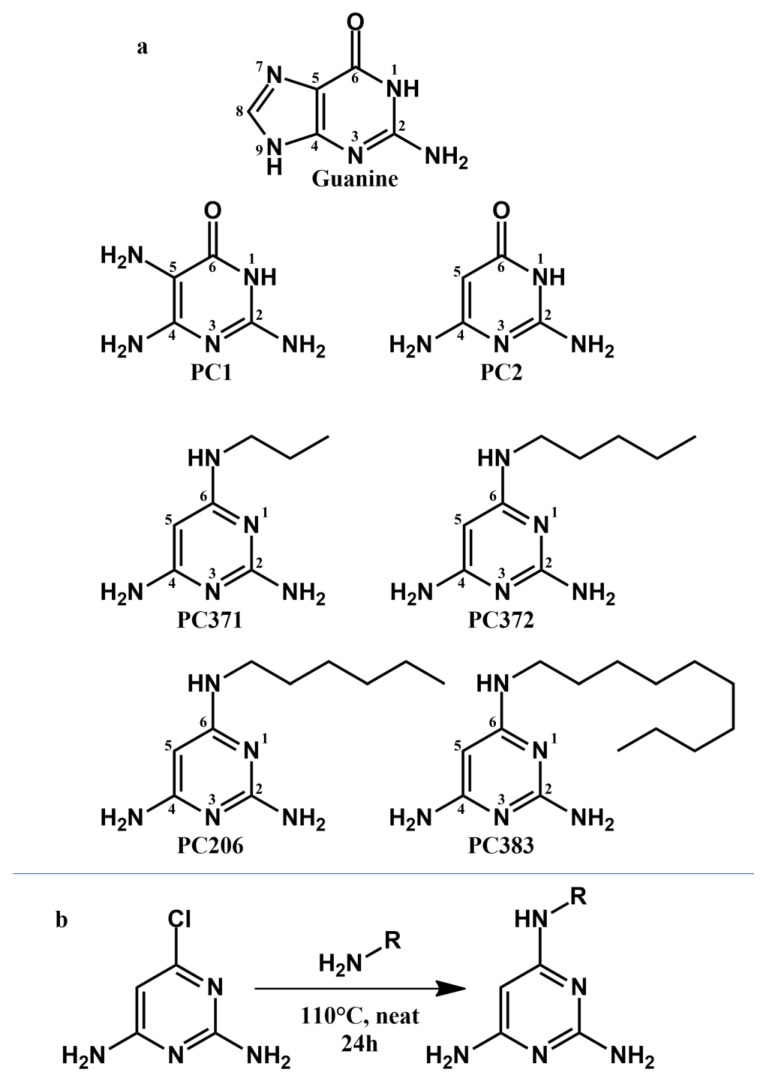
Structures and synthesis of PC371, PC372, PC206, and PC383. (**a**) The structures of pyrimidine compounds differ by the length of the alkyl chain on the amino group at position 6 of the pyrimidine cycle. Guanine, PC1, and PC2 were added as references. (**b**) Pyrimidine compounds were synthesized by the same one-step nucleophilic aromatic substitution protocol.

**Figure 2 antibiotics-12-01344-f002:**
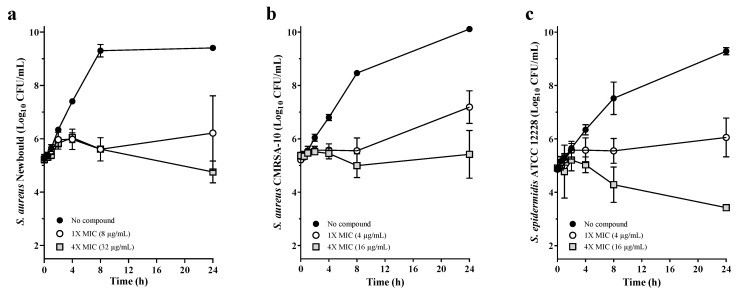
PC206 kill kinetics against staphylococci. The in vitro antibacterial activity of PC206 was evaluated against *S. aureus* Newbould (**a**), *S. aureus* CMRSA-10 (**b**), and *S. epidermidis* ATCC 12228 (**c**). The log_10_ CFU/mL indicated at each time point represents the average of at least three independent experiments. The detection threshold was 2 log_10_ CFU/mL.

**Figure 3 antibiotics-12-01344-f003:**
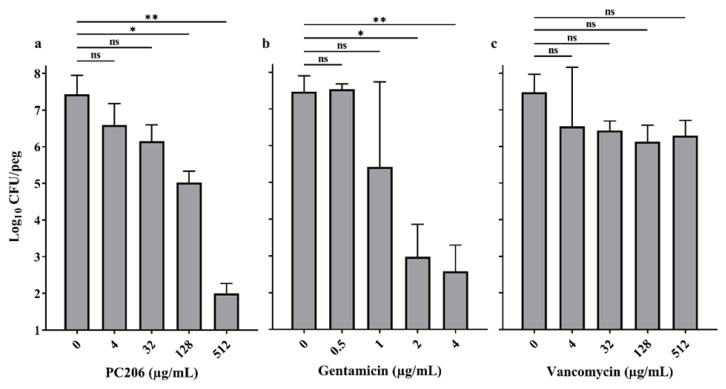
PC206 antibacterial activity against *S. aureus* bovine strain 2117 embedded in preformed biofilm. The activity of PC206 (**a**) and reference antibiotics gentamicin (**b**) and vancomycin (**c**) against 24 h preformed biofilms was assessed. Data were obtained from at least three independent experiments for PC206 and gentamicin and from at least two independent experiments for vancomycin. Statistical analysis was performed using an ordinary nonparametric one-way ANOVA (Kruskal–Wallis test) with multiple comparisons (Dunn’s test), each concentration being compared to the untreated control (ns, not statistically significant; * *p* < 0.05; ** *p* < 0.01).

**Figure 4 antibiotics-12-01344-f004:**
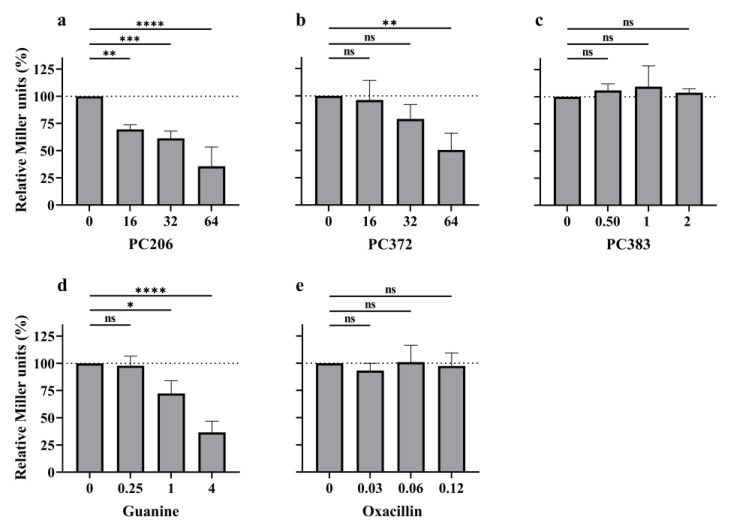
Evaluation of PC206 and analogs in a *lacZ*-gene reporter assay under the control of the guanine riboswitch. The β-galactosidase activity of the *B. subtilis* reporter strain was measured and then normalized by bacterial OD_600_ (Miller units) and relativized to the untreated condition in the presence of several concentrations (µg/mL) of PC206 (**a**), PC372 (**b**), and PC383 (**c**). Guanine (**d**) acts as a positive control; oxacillin (**e**) is a negative control. Data were collected from at least three independent assays. Statistical analysis was performed using an ordinary nonparametric one-way ANOVA (Kruskal–Wallis test) with multiple comparisons (Dunn’s test), each concentration being compared to the untreated control (ns, not statistically significant; * *p* < 0.05; ** *p* < 0.01; *** *p* < 0.001; **** *p* < 0.0001).

**Figure 5 antibiotics-12-01344-f005:**
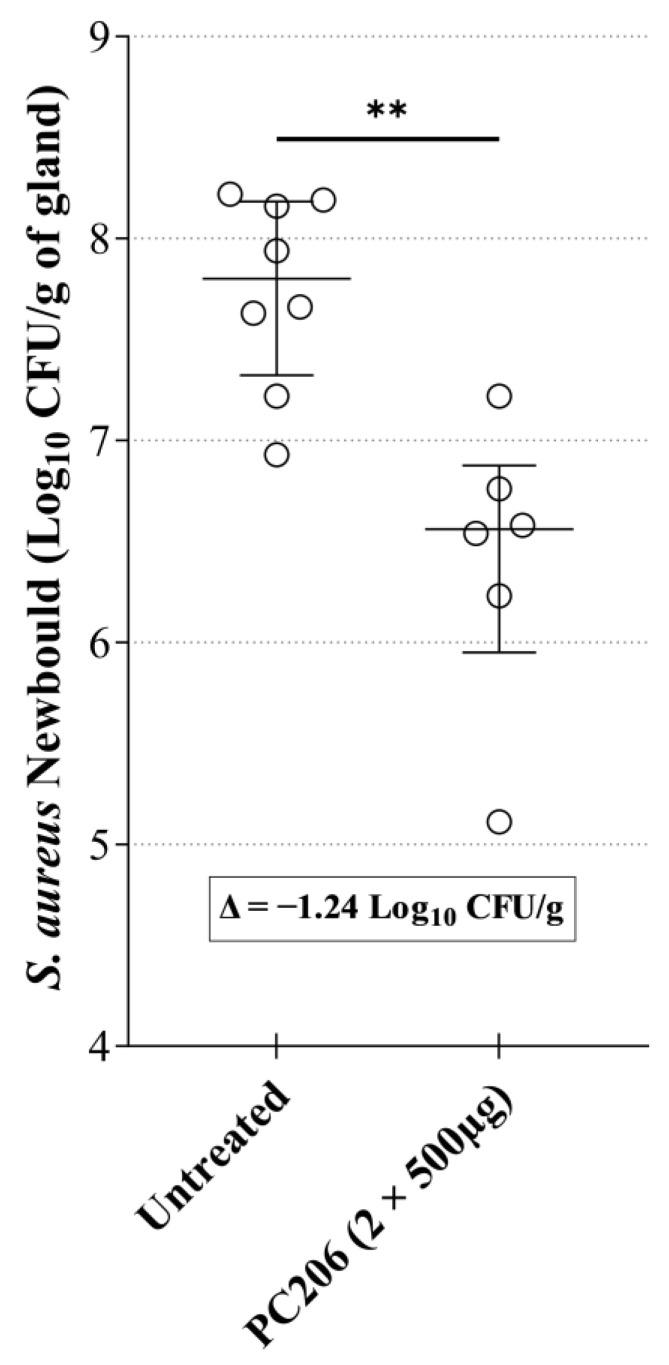
Efficacy of PC206 in the *S. aureus* mouse mastitis model. *S. aureus* Newbould (ATCC 29740) intramammary infections were treated by two doses of 500 µg of PC206 (*n* = 6) and compared to an untreated control (*n* = 8). Mammary glands were harvested 14 h post-infection for determination of the bacterial load (Log_10_ CFU/g of gland). Each circle on the graph represents the bacterial load for each gland. The whiskers show the interquartile range. The horizontal bars indicate the median value for each group. The difference between the medians of the two groups is 1.24 log_10_ CFU/g of gland. Statistical significance was evaluated with a Mann–Whitney test (** *p* ˂ 0.01).

**Figure 6 antibiotics-12-01344-f006:**
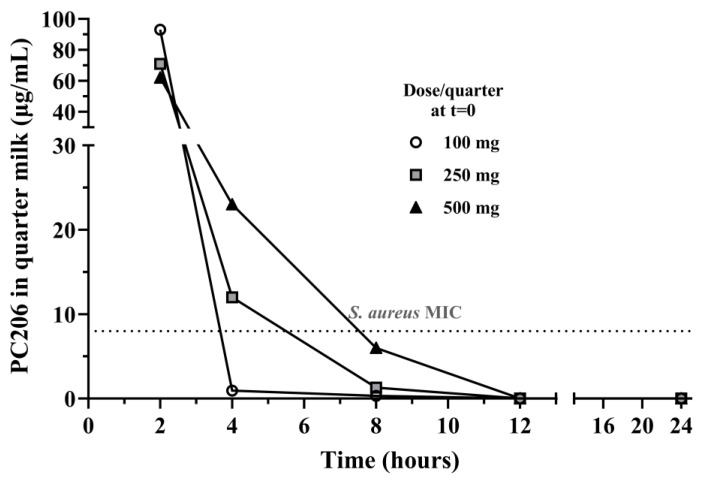
Recovery of PC206 from quarter-milk samples of cows after a single intramammary infusion of 100, 250 or 500 mg. PC206 was extracted and measured as described in the Materials and Methods section. The recovery of PC206 in quarter milk was dose-dependent. The concentration of PC206 remained above the MIC for susceptible *S. aureus* strains found in bovine mastitis (8 µg/mL) for over 2 h for the dose of 100 mg and over 4 h and close to 8 h for the doses of 250 and 500 mg, respectively.

**Table 1 antibiotics-12-01344-t001:** In vitro activity of novel pyrimidine compounds against targeted staphylococci and non-targeted bacterial species.

Bacterial Species	Strain	Broth Microdilution MIC (µg/mL)							
		PC1 ^1^	PC371	PC372	PC206	PC383	VAN ^2^	CAZ ^2^	OXA ^2^
*Staphylococcus aureus*	ATCC 29213	2	>128	16	8	4	0.5	Nd ^3^	0.25
	CMRSA-2	1	>128	16	8	4	1	nd	>64
	CMRSA-10	2	>128	8	4	4	1	nd	64
*Staphylococcus epidermidis*	ATCC 12228	2–4	>128	8	4	4	1	nd	nd
*Enterococcus faecalis*	ATCC 29212	128	>128	>128	128	4	4	nd	nd
*Escherichia coli*	ATCC 35695	>128	>128	>128	>128	16	>128	0.25	nd
	Δ*lptD*4213	>128	>128	>128	>128	8	1	0.03	nd
*Bacillus subtilis*	*xpt-lacZ*	>128	>128	128	128	4	0.25	nd	0.25

^1^ PC1 was tested in the presence of 1.5 mM DTT to prevent dimerization. The MIC of DTT is >75 mM. ^2^ VAN, vancomycin; CAZ, ceftazidime; OXA, oxacillin. ^3^ nd, not done.

**Table 2 antibiotics-12-01344-t002:** Susceptibility (MIC in µg/mL) of several *Staphylococcus* species to PC206.

Species	(N ^1^)	PC206			VAN ^2^			OXA ^2, 3^			PIR ^2^		
		MIC_50_ ^4^	MIC_90_ ^4^	Range ^4^	MIC_50_	MIC_90_	Range	MIC_50_	MIC_90_	Range	MIC_50_	MIC_90_	Range
Bovine *S. aureus*	48	8	16	0.5–64	0.5	0.5	0.25–1	0.12	0.25	0.03–32	0.25	0.5	0.12–1
MRSA	42	8	16	2–128	1	4	0.5–4	32	64	4–64	>64	>64	0.25–64
*S. epidermidis*	10	4	8	2–16	1	1	0.5–1	2	>16	0.03–16	0.12	>16	0.06–16
*S. hominis*	10	8	32	1–32	0.5	0.5	0.25–1	1	>16	0.06–16	1	>16	0.06–16
*S. chromogenes*	5	32	128	16–128	0.5	0.5	0.5	0.12	0.12	0.06–0.12	0.06	1	0.06–2
*S. haemolitycus*	10	64	128	4–128	2	4	0.25–4	>16	>16	0.06–16	0.25	>16	0.06–16
*S. simulans*	5	64	128	64–128	0.5	0.5	0.5	0.06	0.12	0.06–0.12	0.12	0.25	0.12–0.25

^1^ N, number of isolates tested. ^2^ VAN, vancomycin; OXA, oxacillin; PIR, pirlimycin. ^3^ OXA MIC for *Staphylococci* were determined using CAMHB with 2% NaCl. ^4^ MIC_50_, MIC_90_, and MIC ranges are provided in µg/mL.

**Table 3 antibiotics-12-01344-t003:** Relative cytotoxicity (CC_50_ in μM) of PC2 analogs in different assays and cell types.

Compounds	MTT Assay				LDH Assay			
	HepG2 Cells		MAC-T Cells		HepG2 Cells		MAC-T Cells	
	CC_50_	S.I.	CC_50_	S.I.	CC_50_	S.I.	CC_50_	S.I.
PC371	˃600	nc	˃600	nc	˃600	nc	˃600	nc
PC372	341 ± 32	4.2	˃600	nc	394 ± 15	4.8	˃600	nc
PC206	160 ± 31	4.2	317 ± 43	8.3	149 ± 24	3.9	˃600	nc
PC383	11.7 ± 5.8	0.8	13.5 ± 1.5	0.9	6.6 ± 0.7	0.4	23.6 ± 8.4	0.9
NBD-Cl	16.7 ± 7.6	nd	27.6 ± 0.4	nd	9.6 ± 4.5	nd	22.4 ± 0.5	nd

nc: not calculable. nd: not determined. S.I.: selectivity index (CC_50_ values/MIC against *S. aureus* Newbould).

## Data Availability

Additional detailed supporting data are available in the Appendix A.

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
