# Peer review of "Rationally Designed Pyrimidine Compounds: Promising Novel Antibiotics for the Treatment of Staphylococcus aureus-Associated Bovine Mastitis"

_antibiotics, 2023, doi:10.3390/antibiotics12081344_

Round 1

Reviewer 1 Report

The manuscript does not present a conclusion (conclusions), which is necessary for the authors to review. This should briefly state the major findings of the study. 

Author Response

Reviewer 1

Comment 1. The manuscript does not present a conclusion (conclusions), which is necessary for the authors to review. This should briefly state the major findings of the study. 

Response 1. Agreed, the last paragraph of the discussion was modified accordingly (starts at line 424):

In conclusion, although its mode of action remains to be completely elucidated, this study proposes PC206 as the first stable candidate of a novel antibiotic class dedicated to the treatment of S. aureus-induced mastitis in dairy cows. Its potency and specificity were confirmed with MIC assays on large panels of S. aureus isolates from human and animal origins, and against preformed biofilms. Furthermore, its impact on eukaryotic cells was thoroughly examined and deemed suitable. Finally, PC206 safety, effectiveness and pharmacokinetics were evaluated in vivo, including in a mouse mastitis model and in lactating cows.  With one-step synthesis using inexpensive, commercially available material, we expect a low-cost production, which is critical for use in food animal production. This should help reduce broad-spectrum antibiotic overuse in farm animals, allowing to preserve clinically important antimicrobials for human health care.

Reviewer 2 Report

It is an interesting, well-written, and promising manuscript. Here are a few optional suggestions:

1.     Line 38. Use the AMR abbreviation for antimicrobial resistance, and "There" should probably be from a small letter in this line.

2.     As a recommendation, in Tables 1 and 2 MICs can be presented not only in μg/mL, but also in molar (μM), including for antibiotics, thus the difference in the activity of synthesized compounds and vancomycin will be further emphasized because the molecular the mass of vancomycin is much higher.

3.     Figure S4. Sign the figures, which of the figures (a) and which (b).

4.     I think the mechanism of action of the studied compounds should be depicted schematically, and a corresponding figure should be added.

Thank you

Author Response

Reviewer 2

Comment 1. Line 38. Use the AMR abbreviation for antimicrobial resistance, and "There" should probably be from a small letter in this line.

Response 1. The AMR abbreviation has been added to the manuscript when required. On the line 38, “There” has been changed for “there”.

Comment 2. As a recommendation, in Tables 1 and 2 MICs can be presented not only in μg/mL, but also in molar (μM), including for antibiotics, thus the difference in the activity of synthesized compounds and vancomycin will be further emphasized because the molecular the mass of vancomycin is much higher.

Response 2. It is a good suggestion. However, we feel like adding the MIC in molar would make the tables harder to read. Also, traditionally, MIC values are presented in µg/mL (see CLSI, Clinical and Laboratory Standards Institute. Methods for dilution antimicrobial susceptibility tests for bacteria that grow aerobically. 11th Ed. CLSI standard M07 2018 (ISBN 1-56238-836-3 [Print]; ISBN 1-56238-837-1 [Electronic]). CLSI, 950 West Valley Road, suite 2500, PA 19087, USA).

Comment 3. Figure S4. Sign the figures, which of the figures (a) and which (b).

Response 3. Indeed, we have specified which panel is (a), and which is (b).  

Comment 4. I think the mechanism of action of the studied compounds should be depicted schematically, and a corresponding figure should be added.

Response 4. The mechanism of action of our initial lead compound PC1 was thoroughly studied and described in Mulhbacher et al, PlosPathogen (https://doi.org/10.1371/journal.ppat.1000865).  To avoid duplication of illustrations from this article but also since we did not fully establish the mechanism of action of the new analog PC206, we would rather not illustrate a mechanism of action that was not fully determined.

However, we have added a graphical abstract, which explains the rational design of the novel analogs (PC206 and the others), which briefly summarizes the main characteristics of each compound accordingly to their individual structure.

Reviewer 3 Report

The development of new antibiotics for bovine mastitis caused by Staphylococcus aureus is crucial due to its virulence and antibiotic resistance attributes. The newly designed and synthesized PC206 and its analogs, with improved stability and narrow-spectrum activity against S. aureus, demonstrate promising in vitro and in vivo efficacy. These compounds offer a potential solution to address the challenge of antibiotic resistance and provide a promising avenue for treating S. aureus-induced bovine mastitis. The authors need to address some concerns before it is considered for published.

1. Line 91: Please move Figure 1 under this paragraph.

2. Add conclusion part.

3. In the future study, the authors could explore the potential pharmacokinetic and pharmacodynamic properties of PC206, which can help optimize dosing regimens and improve treatment outcomes.

Author Response

Reviewer 3

Comment 1. Line 91: Please move Figure 1 under this paragraph.

Response 1. Figure 1 has been moved under the last paragraph of the introduction.

Comment 2. Add conclusion part.

Response 2. Agreed, the last paragraph of the discussion has been modified accordingly (starts at line 424):

In conclusion, although its mode of action remains to be completely elucidated, this study proposes PC206 as the first stable candidate of a novel antibiotic class dedicated to the treatment of S. aureus-induced mastitis in dairy cows. Its potency and specificity were confirmed with MIC assays on large panels of S. aureus isolates from human and animal origins, and against preformed biofilms. Furthermore, its impact on eukaryotic cells was thoroughly examined and deemed suitable. Finally, PC206 safety, effectiveness and pharmacokinetics were evaluated in vivo, including in a mouse mastitis model and in lactating cows. With one-step synthesis using inexpensive, commercially available material, we expect a low-cost production, which is critical for use in food animal production. This should help reduce broad-spectrum antibiotic overuse in farm animals, allowing to preserve clinically important antimicrobials for human health care.  

Comment 3. In the future study, the authors could explore the potential pharmacokinetic and pharmacodynamic properties of PC206, which can help optimize dosing regimens and improve treatment outcomes.

Response 3. Agreed, the second to last paragraph of the discussion has been modified accordingly (starts at line 407):

The in vivo efficacy of PC206 was demonstrated by using a well-established murine infectious mastitis model [25]. The administration of PC206 reduced viable counts of S. aureus Newbould by 1.24 log10 CFU/g of gland, which was statistically significant. Mouse hepatic and plasmatic stability of PC206 was also assessed and was relatively stable in both environments over the course of the experiment (Supplementary information Figure S3). We then measured the recovery of PC206 in quarter milk of cows after a single intramammary infusion. The concentration of PC206 in the milk exceeded the MIC for S. aureus for nearly 8 h when used at the highest dose (500 mg). In a future study, PC206 in vivo activity, pharmacokinetics and pharmacodynamic properties could be further explored, to optimize dosing regimens and therefore improving treatment outcomes. For example, an appropriate formulation could be tested, to fit dairy farm practices that often require milking every ~12 h. Also note that intramammary infusion is the usual administration procedure for anti-mastitis therapies and that ceftiofur can be used at a dose of 125 mg/quarter every 24 h for up to 8 days or cephapirin at a dose of 250 mg every 12 h. Hence, our infusion procedure and dosing proposition for PC206 is already similar to dairy farm standards.